# UNIFYING DISTRIBUTION ALIGNMENT AS A LOSS FOR IMBALANCED SEMI-SUPERVISED LEARNING

## ABSTRACT

While remarkable progress in imbalanced supervised learning has been made recently, less attention has been given to the setting of imbalanced semi-supervised learning (SSL) where not only is a few labeled data provided, but the underlying data distribution can be severely imbalanced. Recent works require both complicated sampling-based strategies of pseudo-labeled data and distribution alignment of the pseudo-label distribution to accommodate this imbalance. We present a novel approach that relies only on a form of a distribution alignment but no sampling strategy where rather than aligning the pseudo-labels during inference, we move the distribution alignment component into the respective cross entropy loss computations for both the supervised and unsupervised losses. This alignment compensates for both imbalance in the data as well as the eventual distributional shift present during evaluation. Altogether, this provides a single, unified strategy that offers both significantly reduced training requirements and improved performance across both low and richly labeled regimes and over varying degrees of imbalance. In experiments, we validate the efficacy of our method on SSL variants of CIFAR10-LT, CIFAR100-LT, and ImageNet-127. On ImageNet-127, our method shows 1.6% accuracy improvement over the previous best method with 80% training time reduction.

## 1 INTRODUCTION

Semi-supervised learning (SSL) uses a large pool of unlabeled data to learn a classifier despite having access to only a small amount of labeled data. Recently, techniques have been introduced (Berthelot et al., 2019; 2020; Sohn et al., 2020) which simplify the process while at the same time pushing performance to new levels. However, these approaches have focused on the cases where the class distributions are balanced for both the labeled and unlabeled data.

At the same time, work within the *supervised learning* community has shown renewed focus on imbalanced or sometimes long-tailed learning — owing to the fact that most data in the real world is not well-balanced. A variety of methodologies for this setting have been introduced (Kang et al., 2020; Menon et al., 2021; Ren et al., 2020; Hong et al., 2021). For instance, many have observed the bias that ordinary supervised learning techniques suffer from — favoring *head classes* over the less numerous *tail classes*. Kang et al. (2020) show that softmax-based classifiers often produce classification weights which correlate with class frequency and thus reduces the class-balanced performance of the model. Resampling strategies (Chawla et al., 2002; He & Garcia, 2009; Buda et al., 2018; Byrd & Lipton, 2019) (which sample from the pool based on the desired distribution) are often effective as well. By shifting from sampling the data distribution for the majority of training to a more class-balanced regime at the end, one can learn a good representation while mitigating the aforementioned bias at the final classification layer. In addition, a distributional shift (Hong et al., 2021) can be observed within existing protocols where training occurs on an imbalanced dataset yet evaluation is done with respect to a balanced one.

In this work, we study the combined setting of semi-supervised and imbalanced learning. In particular, we consider both FixMatch (Sohn et al., 2020) and MixMatch (Berthelot et al., 2019) as base semi-supervised learners. Both employ two losses: a cross entropy loss on the labeled data, and an unsupervised loss that relies on consistency between the classifier outputs among augmented versions of unlabeled examples.

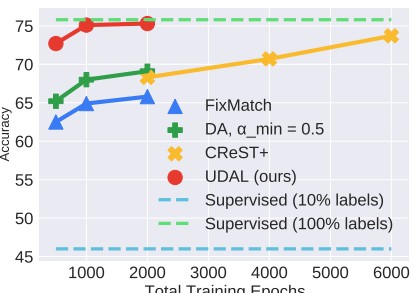

Figure 1: **Large-scale performance on ImageNet-127 with 10% of labels available**. The proposed UDAL trained for only $1/6$ of the epochs of CReST+ (Wei et al., 2021) further closes the gap in accuracy to a fully supervised baseline and heavily improves on a lower bound supervised baseline trained on 10% of the labels.

Therefore, vulnerability to bias from the imbalance can happen in three ways: the supervised loss itself, the quality of pseudo-labels derived from the classifier on unlabeled examples, and the pseudo-labels themselves can bias the classifier *even if they are perfectly predicted*. Furthermore, confirmation bias within semi-supervised learning is already a worrisome factor even without any imbalance between the classes (Arazo et al., 2020). Within the balanced setting, the ReMixMatch (Berthelot et al., 2020) approach to semi-supervised learning introduces strong regularization through *distribution alignment* (Bridle et al., 1992) (i.e. modifying the prediction by the ratio of the desired distribution to model distribution) to help mitigate this. Recently CReST (Wei et al., 2021) has shown that distribution alignment also confers benefits in the imbalanced setting and aims to *progressively rebalance* the distribution of pseudo-labels. Similar to resampling approaches (Kang et al., 2020), imbalanced learning considers a shift from random sampling to class-balanced sampling, CReST attempts to align the pseudo-labels themselves to a more balanced distribution as training progresses.

However, distribution alignment is not the only technique that CReST relies upon to achieve good performance. In addition, CReST requires a generational approach to self-training which accumulates a relatively balanced subset of confident pseudo-labels to augment the *labeled* set with as a form of self-training. As the generations proceed, this subset is re-sampled to become more and more balanced. Each generation *re-initializes* the classifier's network, and therefore, the only "state" retained is through these accumulated pseudo-labels (now treated as ordinary supervised labels). This process can be extremely costly with respect to training time, and we hypothesize that it is not optimal because it fails to directly address the imbalance in the labeled data. Instead, we seek a simpler solution to imbalanced semi-supervised learning through distribution alignment alone.

We ask the question: *is this disjoint methodology truly necessary?* Can a single, central approach be devised to address imbalance in semi-supervised learning? In this work, as our **contributions**, we identify an affirmative answer to this question by connecting the ideas of progressive distribution alignment from Berthelot et al. (2020); Wei et al. (2021) and the method of logit adjustment from fully supervised, imbalanced learning (Menon et al., 2021; Ren et al., 2020; Hong et al., 2021). Furthermore, this approach can be implemented with only a few lines of code, has significantly reduced training time requirements, and generally outperforms previous work. Finally, it shows *significantly better* performance characteristics as more labeled data becomes available and readily scales to larger datasets — achieving a 1.6% increase in accuracy compared to the best existing method on ImageNet-127.

## 2 Prerequisites

We present background information on both problem settings. First, we formally define the problem setting of Imbalanced Semi-Supervised Learning (SSL). Second, we outline the idea of distribution alignment (Berthelot et al., 2020; Wei et al., 2021) to improve pseudo-label quality within both the balanced and imbalanced settings of SSL. We revisit a recent method, CReST (Wei et al., 2021), which also attempts to address imbalanced semi-supervised learning.

### 2.1 Class-imbalanced Semi-Supervised Learning

Semi-supervised learning relies on two sources of data: a *labeled* set $\mathcal{X} = \{(x_i, y_i)\}_{i=1}^{N}$ where each $x_i$ is a training example and $y_i$ is the corresponding target. Since classification is the focus of this work, we consider $y_i$ as a class label within $\mathcal{C} = \{1, \ldots, C\}$ with a total number of $C$ classes. In imbalanced learning, we expect varying numbers of training examples across classes. Therefore,

we denote the number of examples in our labeled set corresponding to class $c \in \mathcal{C}$ as $N_c$ such that $\sum_{c=1}^{C} N_c = N$. We assume that the classes are ordered with respect to frequencies and in a descending manner i.e. $N_c \geq N_{c+1}$. It is often useful to characterize the degree of imbalance by the ratio $N_1/N_C$, and we refer to this as the *imbalance ratio* of the dataset. We use $p_{\text{data}}(y)$ and $q(y)$ to denote the marginal distributions of the data and the model. When there is no ambiguity, we drop $y$ as $p_{\text{data}}$ and $q$ to simplify presentations.

Additionally, we have an *unlabeled* set of examples $\mathcal{U} = \{u_i\}_{i=1}^{M}$ for which we have no corresponding target. While we expect that this set is also imbalanced, we additionally make the common assumption (Wei et al., 2021) that it follows the same class distribution and thus shares the same imbalance (ratio) as the labeled set. Finally, an important measure $\beta = \frac{N}{N+M}$ considers the percentage of overall examples that are labeled.

## 2.2 Distribution Alignment

Distribution alignment (DA) was re-introduced in the setting of semi-supervised learning within ReMixMatch (Berthelot et al., 2020). To mitigate the tendencies of semi-supervised learning to suffer from confirmation bias, regularization can be added to the pseudo-label inference step. In particular, if we assume the labeled and unlabeled data both come from the same distribution $p_{\text{data}}$ (although, we do not know *particular* labels from the unlabeled set), we would expect that our model should produce pseudo-labels that follow the same distribution. This marginal distribution of the model $q(y)$ can be estimated by moving average, which we denote as $\hat{q}(y)$ or $\hat{q}$, as in Berthelot et al. (2020). If we denote our current model's predictions on unlabeled examples as $q(y|x_u)$, these predictions can be re-scaled through dividing by $\hat{q}$ and multiplying by $p_{\text{data}}$. After normalization, we have:

$$\tilde{q}(y|x) = \text{Normalize}\left( q(y|x) \, \frac{p_{\text{data}}}{\hat{q}} \right). \tag{1}$$

In equation 1, we assume element-wise operations beween $q(y|x)$, $p_{\text{data}}$ and $\hat{q}$. Normalize($p$) ensures $p$ as a probability distribution which sums to 1.

As noted in Wei et al. (2021), it is not always optimal to align the predictions directly to $p_{\text{data}}$ when $p_{\text{data}}$ is imbalanced. Rather, a smoothed form which (elementwise) exponentiates the distribution by a factor of $\alpha$ before normalization:

$$\tilde{p}_\alpha = \text{Normalize}\left( p_{\text{data}}^{\alpha} \right), \; 0 \leq \alpha \leq 1. \tag{2}$$

is used instead of $p_{\text{data}}$ and found to both regularize the predictions as well as combat bias. As $\alpha \to 0$, this approaches an alignment against a more uniform distribution.

## 2.3 Logit Adjustment

While DA is clearly applicable to pseudo-label *inference* during training, it has no direct effect on the labeled portion. Since a semi-supervised approach relies on both labeled and unlabeled losses, it is critical to address the problem of imbalance at the supervised level as well. For this, we examine a popular technique within supervised learning, often known as logit adjustment (Menon et al., 2021), balanced softmax (Ren et al., 2020), or LADE (Hong et al., 2021). These methods modify the *loss computation* to compensate for the class imbalance found in the data distribution. Notably, when a data distribution is class-imbalanced, we attempt to minimize the classification loss with respect to this data distribution. *However*, at evaluation time, we either evaluate on a class-balanced dataset or produce a class-balanced error by averaging the per-class accuracies. This is a shift in distribution which can cause poor performance. Therefore, this shift is integrated into the cross entropy loss:

$$\mathcal{L}_{LA}\left(y, f(x)\right) = \mathcal{L}_{CE}\big(y, f(x) + \log p_{\text{data}} - \log\left(\text{Unif}(C)\right)\big) \equiv \mathcal{L}_{CE}\big(y, f(x) + \log p_{\text{data}}\big) \tag{3}$$

where Unif($C$) is the discrete uniform distribution over $C$ classes, $y$ is the true label of $x$, $f(x)$ is the vector-valued output of the classifier, and $p_{\text{data}}$ is the marginal class distribution of the data as a vector. As elaborated within Menon et al. (2021), this has the effect that instead of optimizing $f(x)$ directly, we optimize $h(x) = f(x) + \log p_{\text{data}}$ which aligns the source distribution of $f(x)$ correctly to the uniform class distribution seen during evaluation. Menon et al. (2021) also discusses an inference time procedure which attempts to account for this shift without modifications to the training procedure. Since this is "for free", we include results combined with it in Table 1 as "LA (Inf)" for the inference time procedure.

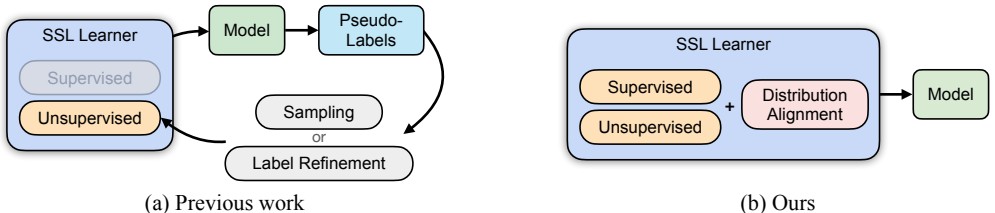

(a) Previous work                                                           (b) Ours

Figure 2: Given an off-the-shelf Semi-Supervised Learning (SSL) learner, previous works address the data imbalanced issue on the unsupervised loss by *iterative* (a) class-rebalancing sampling (Wei et al., 2021) or pseudo-label refinement (Kim et al., 2020). In this work, we propose to tackle the imbalance issue on (b) both supervised and unsupervised losses by *directly* performing distribution-aligned learning.

## 2.4 CReST

We briefly examine CReST (Wei et al., 2021), an approach to imbalanced semi-supervised learning using FixMatch and MixMatch as a base semi-supervised learner. They re-introduce a generational self-training approach where after each generation (usually 64 epochs), confidently pseudo-labeled examples from the unlabeled set $\mathcal{U}$ are added to $\mathcal{X}$. Ordinarily, this would *exacerbate* class imbalance. Wei et al. (2021) introduce a re-sampling strategy according to a more balanced form of the marginal class distribution of the data to overcome. However, it requires to *reinitialize and train the network from scratch* after each generation.

Additionally, for optimal performance, CReST requires distribution alignment to improve pseudo-label quality. Specifically, as defined in Section 2.2, a *schedule* for $\alpha$ is chosen so that the strength of re-balancing can be altered over the course of training. Given a hyperparameter $\alpha_{\min}$ which defines the minimal value $\alpha$ should take (corresponding to the largest re-balancing of the data distribution), they choose a linear schedule such that for generation iteration $t$:

$$\alpha_t = 1.0 - (1.0 - \alpha_{\min}) \frac{t}{T} \tag{4}$$

where $T$ is the number of generations over the course of training. This is referred to as *progressive distribution alignment* and is essential to strong performance.

## 3 PROPOSED METHOD

We describe our approach, Unifying Distribution Alignment as a Loss (UDAL). First, we connect Sections 2.2 (distribution alignment) and 2.3 (logit adjustment). Then, we examine how this allows us to apply the same mitigation to both the supervised and unsupervised components of modern consistency-based SSL methods (Berthelot et al., 2019; Sohn et al., 2020) – providing a unified manner in which imbalance can be addressed with respect to labeled and unlabeled data. Figure 2 shows the direct comparison between the proposed UDAL and previous work.

### 3.1 A UNIFIED APPROACH

We argue that the essence of progressive distribution alignment as seen in Sections 2.2 and 2.4 *is sufficient* for strong performance in imbalanced semi-supervised learning if viewed through the lens of logit adjustment from Section 2.3. For the sake of presentation, we will use FixMatch (Sohn et al., 2020) as a base semi-supervised learner, but our method is equally applicable to other methods, including MixMatch (Berthelot et al., 2019). FixMatch consists of a cross entropy loss on the labeled data and a cross entropy loss on unlabeled data with respect to inferred pseudo-labels:

$$\mathcal{L}_{\text{FixMatch}} = \mathbb{E}_{x,y \sim p_{\text{label}}} \left[ \mathcal{L}_{\text{CE}}(y, f(x)) \right] + \mathbb{E}_{u \sim p_{\text{unlab}}} \left[ \mathcal{L}_{\text{CE}} \left( PL(f(u^{\text{weak}})), f(u^{\text{strong}}) \right) \right] \tag{5}$$

where $f(x)$ are the network's outputs, $p_{\text{label}}$ is labeled data distribution, $p_{\text{unlab}}$ is the unlabeled data distribution, $PL$ produces hard pseudo-labels for confident predictions, while $u^{\text{strong}}$ and $u^{\text{weak}}$ are strong and weakly augmented versions of the unlabeled example $u$, respectively.

As described in Section 2.2 and 2.4, applying progressive distribution alignment modifies predictions of unlabeled data by aligning them to a *target distribution*. While this aligns the predictions

of the unsupervised branch, we must also align the supervised branch. However, distribution alignment modifies the inference process and is unable to affect the known (true) labels of the supervised branch. Nonetheless, logit adjustment (Menon et al., 2021) demonstrates that logit adjustment can be used to align a supervised classifier to the uniform distribution $\text{Unif}(C)$. Therefore, we examine whether it can simply replace the usage of distribution alignment during pseudo-label inference. Since we are interested in aligning to an arbitrary target distribution like in DA, we consider a generalization of Equation 3 that aligns to $\tilde{p}_{\alpha_t}$ instead of $\text{Unif}(C)$:

$$\mathcal{L}_{\text{CE}}\left(y, f(x) + \log\left(p_{\text{data}}\right) - \log\left(\tilde{p}_{\alpha_t}\right)\right) \tag{6}$$

Defining $h(x) = f(x) + \log\frac{p_{\text{data}}}{\tilde{p}_{\alpha_t}}$ and solving for $f(x)$, this implies that we're optimizing:

$$f(x) = h(x) + \log\left(\frac{\tilde{p}_{\alpha_t}}{p_{\text{data}}}\right) \tag{7}$$

or rather, in the softmax space:

$$\text{Softmax}(f(x)) = \text{Softmax}\left(h(x)\frac{\tilde{p}_{\alpha_t}}{p_{\text{data}}}\right) \tag{8}$$

This is exactly the form of applying distribution alignment to $f(x)$ using $\tilde{p}_{\alpha_t}$. Therefore, we hypothesize that not only can we align the supervised branch in this manner, but that we can also apply this exact same form to the unsupervised branch:

$$\mathcal{L}_{\text{CE}}\left(y, f(x) + \log\left(\frac{\hat{q}}{\tilde{p}_{\alpha_t}}\right)\right) \tag{9}$$

where we divide by the estimated model's class distribution $\hat{q}$ through moving average, rather than $p_{\text{data}}$ since the unsupervised branch is learned from pseudo-labels drawn directly from the model rather than the data distribution. *This allows us to use the same, unified approach for each branch* by simply replacing the distribution we are aligning from: the marginal class distribution of the data for the supervised branch and the moving average of pseudo-labels for the unsupervised branch. Altogether, this results in a loss:

$$\begin{aligned}\mathcal{L}_{\text{UDAL}} = {} & \mathbb{E}_{x,y\sim p_{\text{label}}}\left[\mathcal{L}_{\text{CE}}(y, f(x) + \log\left(p_{\text{data}}\right) - \log\left(\tilde{p}_{\alpha_t}\right))\right] \\ & + \mathbb{E}_{u\sim p_{\text{unlab}}}\left[\mathcal{L}_{\text{CE}}\left(PL(f(u^{\text{weak}}), f(u^{\text{strong}}) + \log\left(\hat{q}\right) - \log\left(\tilde{p}_{\alpha_t}\right))\right)\right]\end{aligned}$$

where $\alpha_t$ is generalized from Equation 4 as:

$$\alpha_t = 1.0 - (1.0 - \alpha_{\min})\left(\frac{t}{T}\right)^k \tag{10}$$

to allow for a rate of alignment $k$ and where $\tilde{p}_{\alpha_t}$ is computed as in Equation 2.

We find that sharing this exact same form between branches has performance benefits as outlined in Section 4.5.1. At the same time, unlike distribution alignment, this form does not directly modify the PL inference procedure. One might wonder whether a progressive form of alignment is even necessary for the supervised branch. We observe heavily degraded performance when progressive alignment is not used in Table 1 under "LA (Sup)".

For MixMatch, an $L_2$ loss function is used between predictions and *soft targets*. We find that applying the same adjustment procedure as in FixMatch but under the $L_2$ loss continues to work as long as $\alpha_{\min}$ is changed to compensate.

We provide pseudocode for this approach in Algorithm 1. This constitutes *only a few lines of code* with no alterations to the training scheme or time compared to the base FixMatch/MixMatch learner. Additionally, no resampling is required.

## 4 EXPERIMENTS

We follow the experimental settings outlined in CReST (Wei et al., 2021). As done in Wei et al. (2021), we make the assumption that the marginal class distributions of labeled and unlabeled data are equal. We show the performance of UDAL over long-tailed versions of CIFAR10 and CIFAR100, as well as CIFAR10 in the "DARP" (Kim et al., 2020) regime. Finally, we perform large-scale experiments on the naturally long-tailed ImageNet-127 dataset.

---

**Algorithm 1** UDAL Pseudocode, TensorFlow-ish

---

```
# p_data: class distribution of labeled data
# p_model: moving average of model's predictions on unlabeled data
# current_epoch (g): current epoch of training (out of max_epoch total)
# k: rate at which a_min is approached
# a_min: minimum value of alpha for distribution alignment
def compute_adjustment_dist(current_dist):
    factor = 1.0 - (1.0 - a_min) * (current_epoch / max_epoch) ** k
    # normalize ensures the argument sums to 1
    target_dist = normalize(p_data ** factor)
    return current_dist / (target_dist + 1e-9)

# compute supervised (labeled) loss on examples x_l
# f: classifier
# y_l: true labels
loss_l = SoftmaxCE(y_l, f(x_l) + log(compute_adjustment_dist(p_data)))

# compute unsupervised (unlabeled) loss on examples x_u
# y_u: PLs predicted from weakly-augmented x_u after confidence thresholding
loss_u = SoftmaxCE(y_u, f(x_s) + log(compute_adjustment_dist(p_model)))
```

---

| Method | $\gamma = 50$ | $\gamma = 100$ | $\gamma = 200$ |
|---|---|---|---|
| MixMatch Berthelot et al. (2019) | $69.1_{\pm 1.18}$ | $60.4_{\pm 2.24}$ | $54.5_{\pm 1.87}$ |
| w/ CReST+ Wei et al. (2021) | $76.7_{\pm 0.35}$ | $66.1_{\pm 0.79}$ | $57.6_{\pm 1.30}$ |
| w/ UDAL (ours) | $\mathbf{77.8}_{\pm 0.88}$ | $\mathbf{68.4}_{\pm 1.48}$ | $\mathbf{58.6}_{\pm 1.10}$ |
| FixMatch (Sohn et al., 2020) | $80.1_{\pm 0.44}$ | $67.3_{\pm 1.19}$ | $59.7_{\pm 0.63}$ |
| w/ DA (Berthelot et al., 2020) ($\alpha_{\min} = 0.5$) | $82.4_{\pm 0.33}$ | $73.6_{\pm 0.63}$ | $63.7_{\pm 1.17}$ |
| w/ DA ($\alpha_{\min} = 0.5$) + LA (Sup) | $83.5_{\pm 0.19}$ | $75.7_{\pm 1.56}$ | $65.7_{\pm 1.87}$ |
| w/ LA (Inf) (Menon et al., 2021) | $83.2_{\pm 0.87}$ | $70.4_{\pm 2.90}$ | $62.4_{\pm 1.24}$ |
| w/ CReST+ (Wei et al., 2021) | $84.2_{\pm 0.39}$ | $78.1_{\pm 0.84}$ | $67.7_{\pm 1.39}$ |
| w/ CReST+ & LA (Inf) (Wei et al., 2021) | $85.6_{\pm 0.36}$ | $81.2_{\pm 0.70}$ | $71.9_{\pm 2.24}$ |
| w/ UDAL (ours) | $85.3_{\pm 0.34}$ | $80.2_{\pm 0.59}$ | $68.6_{\pm 1.32}$ |
| w/ UDAL & LA (Inf) (ours) | $\mathbf{86.3}_{\pm 0.37}$ | $\mathbf{82.1}_{\pm 0.37}$ | $\mathbf{72.9}_{\pm 1.21}$ |

Table 1: We compare baselines on CIFAR10-LT at $\beta = 10\%$, from the simplest approaches that involve no accommodations for class imbalance to stronger baselines which attempt to address it. We note that we include two types of logit adjustment (Menon et al., 2021) (LA) here: one that involves modifying the loss to the supervised branch ("LA Sup") to account for distributional shift and another that does not modify training and applies an adjustment factor at inference time ("LA (Inf)").

## 4.1 CIFAR-LT

## 4.2 DATASET CREATION

CIFAR10LT and CIFAR100-LT (Cao et al., 2019; Cui et al., 2019) are modifications of CIFAR10/CIFAR100 with a long-tailed (Zipfian) distribution. Given a desired imbalance ratio $\gamma$ and some class ordering $C_i$, $1 \leq i \leq C$, we sample $N_i$ examples from the dataset for class $C_i$ according to $N_i = N_1 * \gamma^{\frac{C_i - 1}{C - 1}}$. For CIFAR10, $N_1 = 5000$, $C = 10$ while $N_1 = 500$ and $C = 100$ for CIFAR100. In order to create suitable splits for semi-supervised learning, labeled subsets are randomly sampled according to $\beta = 10\%$ and $30\%$. Imbalance ratios of $\gamma \in \{50, 100, 200\}$ are explored for CIFAR10-LT while $\gamma \in \{50, 100\}$ for CIFAR100-LT. We evaluate with respect to the original, balanced test set which ensures that we can use ordinary accuracy metrics.

## 4.3 TRAINING

We follow the same model guidelines in CReST (Wei et al., 2021). Therefore, we use Wide ResNet-28 (with width 2) as a backbone. FixMatch (Sohn et al., 2020) and MixMatch (Berthelot et al., 2019) are used as base semi-supervised learners. With respect to FixMatch, the unlabeled ratio (7) and confidence threshold (0.95) are untouched from the original paper's settings. The same cosine learning rate schedule is adopted as in Wei et al. (2021). Notably different from (Wei et al., 2021), however, is our ability to use vastly shorter training times. Generally, we find only 64 epochs ($2^{16}$ iterations at a batch size of 64) is necessary to reach optimal performance, and therefore all CIFAR

models are trained for 64 epochs — constituting *a 5x reduction in training time* compared to CReST. Like CReST (Wei et al., 2021), we find it useful to train MixMatch models slightly longer and use a less aggressive $\alpha_{\min}$, therefore, MixMatch models are trained for 128 epochs and use $\alpha_{\min} = 0.5$.

Apart from those of the base semi-supervised learner, our method introduces only two hyperparameters: $\alpha_{\min}$ which controls the final strength of the re-balancing for $\tilde{p}_{\alpha_t}$ and $k$, which controls the rate at which we approach $\alpha_{\min}$. We use $\alpha_{\min} = 0.10$ and $k = 2$ which allows most of training to be aligned to $p_{\text{data}}$ and spend the very last stages of training aligning to a relatively balanced class distribution. This is supported by other work (Kang et al., 2020) which empirically finds that a when training long-tailed, fully-supervised models, the bulk of training should be done with respect to random sampling, and only a small amount of time of class-balanced sampling is necessary near the end of training. We provide an ablation of these hyperparameters in Section 4.5.2.

Training is carried out over 5 random folds of the data. We report final test accuracy along with standard deviation using the exponential moving average of the model's parameters. We use the same underlying codebase, written in TensorFlow (Abadi et al., 2015), and data splits as CReST as recommended in (Oliver et al., 2018) to prevent framework or dataset dependent uncertainty in performance.

## 4.4 RESULTS

### 4.4.1 CIFAR

CIFAR10-LT and CIFAR100-LT results are summarized in Table 2. Across the board, we find UDAL is competitive with CReST+ (Wei et al., 2021) and is only outperformed in a single setting. Notably, we find that CReST+ *especially struggles* in the more label rich regimes. In particular, despite a heavy imbalance of $\gamma \in \{100, 200\}$, UDAL significantly outpeforms CReST+ when $\beta = 30\%$ for both CIFAR10-LT (up to a 4 point increase) and CIFAR100-LT (up to a 1.7 point increase). We hypothesize that CReST+ is unable to efficiently use more labeled data since it only weakly addresses imbalance in the supervised branch. The imbalance in the supervised branch can only be indirectly alleviated by adding pseudo-labeled to the dataset under some amount of resampling. While this could have a high impact when there is scarce labeled data to begin with, it appears to have much less of an effect when more labeles are available. UDAL, however, directly aligns the supervised branch to a more balanced distribution over the course of training. Not only would this improve the balanced performance of the classifier in the fully supervised setting, but it additionally aids the feedback loop to provide more balanced pseudo-labels in the unsupervised branch.

| | CIFAR10-LT | | | | | | CIFAR100-LT | | | |
| | $\beta = 10\%$ | | | $\beta = 30\%$ | | | $\beta = 10\%$ | | $\beta = 30\%$ | |
| Method | $\gamma = 50$ | $\gamma = 100$ | $\gamma = 200$ | $\gamma = 50$ | $\gamma = 100$ | $\gamma = 200$ | $\gamma = 50$ | $\gamma = 100$ | $\gamma = 50$ | $\gamma = 100$ |
|---|---|---|---|---|---|---|---|---|---|---|
| FixMatch (Sohn et al., 2020) | $79.4_{\pm 0.65}$ | $66.3_{\pm 1.74}$ | $59.7_{\pm 0.74}$ | $81.9_{\pm 0.30}$ | $73.1_{\pm 0.58}$ | $64.7_{\pm 0.69}$ | $33.7_{\pm 0.94}$ | $28.3_{\pm 0.66}$ | $43.1_{\pm 0.24}$ | $38.6_{\pm 0.45}$ |
| w/ CReST+ (Wei et al., 2021) | $84.2_{\pm 0.39}$ | $78.1_{\pm 0.84}$ | $67.7_{\pm 1.39}$ | $84.9_{\pm 0.27}$ | $79.2_{\pm 0.20}$ | $70.5_{\pm 0.56}$ | $38.8_{\pm 1.03}$ | $\mathbf{34.6}_{\pm 0.74}$ | $46.7_{\pm 0.34}$ | $42.0_{\pm 0.44}$ |
| w/ UDAL (ours) | $\mathbf{85.3}_{\pm 0.34}$ | $\mathbf{80.2}_{\pm 0.59}$ | $\mathbf{68.6}_{\pm 1.32}$ | $\mathbf{86.7}_{\pm 0.34}$ | $\mathbf{82.4}_{\pm 0.43}$ | $\mathbf{74.5}_{\pm 1.13}$ | $\mathbf{39.8}_{\pm 0.88}$ | $34.3_{\pm 0.85}$ | $\mathbf{48.0}_{\pm 0.56}$ | $\mathbf{43.7}_{\pm 0.41}$ |

Table 2: Classification accuracy (%) over CIFAR10-LT and CIFAR100-LT under a variety of label fractions $\beta$ and imbalance ratios $\gamma$, each averaged over 5 folds.

**DARP** We show results within the DARP (Kim et al., 2020) setting in Table 3. While DARP also produces an imbalanced dataset from CIFAR10, it has slight differences. Whereas the CReST (Wei et al., 2021) setting assigns every example from the original CIFAR dataset to *either* the labeled or unlabeled dataset, DARP uses an unlabeled to labeled ratio of 2:1, which may or may not utilize all data points. One can consider the DARP setting as a $\beta = 33\%$ but with only 95% of the original dataset.

Nonetheless, UDAL continues to significantly outperforms CReST+ in all settings and shows the largest gains of all in the most imbalanced settings — capable of taking advantage of the increased amount of labeled data.

### 4.4.2 IMAGENET-127

We additionally provide experimental results on ImageNet-127 in Table 4. As presented in (Wei et al., 2021), ImageNet-127 is a coarser version of ImageNet (Krizhevsky et al., 2012), containing

| Method | $\gamma = 50$ | $\gamma = 100$ | $\gamma = 150$ |
|---|---|---|---|
| FixMatch (Sohn et al., 2020) | $79.2_{\pm 0.33}$ | $71.5_{\pm 0.72}$ | $68.4_{\pm 0.15}$ |
| w/ DARP (Kim et al., 2020) | $81.8_{\pm 0.24}$ | $75.5_{\pm 0.05}$ | $70.4_{\pm 0.25}$ |
| w/ CReST+ (Wei et al., 2021) | $83.9_{\pm 0.14}$ | $77.4_{\pm 0.36}$ | $72.8_{\pm 0.58}$ |
| w/ UDAL (ours) | $\mathbf{86.5}_{\pm 0.29}$ | $\mathbf{81.4}_{\pm 0.39}$ | $\mathbf{77.9}_{\pm 0.33}$ |

Table 3: Classification accuracy (%) under DARP's protocol (Kim et al., 2020) for CIFAR10.

the same number of examples but with 127 class groupings rather than 1000. This grouping results in an imbalance ratio of $\gamma = 286$. We conduct experiments with $\beta = 10\%$ along with all base training settings identical to CReST+, with exception to the duration of training, where we observe that only 1000 epochs (1/6 the total training time of CReST+) is necessary for convergence (Figure 1). For UDAL, we continue to find $k = 2$ to be optimal, however, a less aggressive $\alpha_{\min} \in [0.5, 0.6]$ was found to be more effective in this setting. Overall, we find UDAL provides quite healthy increases in performance over CReST and is only 0.7 absolute points worse than a fully supervised baseline with access to 100% of the labels.

| Method | 1000 | 2000 | 4000 | 6000 |
|---|---|---|---|---|
| | | Epochs | | |
| Supervised (100% labels) | | 75.8 | | |
| Supervised (10% labels) | | 46.0 | | |
| FixMatch (10% labels) | 64.9 | 65.8 | - | - |
| w/ DA ($\alpha_{\min} = 0.5$) | 68.0 | 69.1 | - | - |
| w/ CReST+ | - | 68.3 | 70.7 | 73.7 |
| w/ UDAL (ours, $\alpha_{\min} = 0.5$) | 74.5 | 74.3 | - | - |
| w/ UDAL (ours, $\alpha_{\min} = 0.55$) | $\mathbf{75.1}$ | 74.8 | - | - |
| w/ UDAL (ours, $\alpha_{\min} = 0.6$) | 73.2 | $\mathbf{75.3}$ | - | - |

Table 4: Evaluating the proposed method on ImageNet127 with imbalance factor $\gamma = 286$ where $\beta = 10\%$ samples are labeled. Supervised models are trained for 300 epochs with 100% labeled data or equivalently 3000 epochs with 10% labeled data.

## 4.5 ABLATION STUDIES

We carry out ablation studies for our method using the CIFAR10-LT dataset with $\gamma = 100$ and $\beta = 10\%$. We plot all graphs with respect to the same scale so that they may be readily compared.

### 4.5.1 REPLACING CReST'S SAMPLING PROCESS

We explore whether it would be sufficient to simply replace CReST's generational sampling process with the supervised component of our method. The essential difference between this and our method would be the usage of distribution alignment within the *pseudo-label inference* (i.e., this affects the pseudo-labeling process directly) versus enforcing a modified loss computation without any change to the pseudo-label inference process. We consider a variety of schedules in Figure 3a. While this approach performs well, and can even slightly outperforms CReST+, it still falls short of UDAL, which uses the same approach for both supervised and unsupervised losses — without the need to alter the pseudo-label inference process.

### 4.5.2 SCHEDULE PARAMETERS

Since UDAL uses a form of progressive distribution alignment in a similar way to CReST, it introduces two hyperparameters: $k$ and $\alpha_{\min}$ which dictates the extent and rate at which the alignment approaches the uniform distribution. An ablation of these values on CIFAR10-LT is shown in Figure 3b which supports that a quadratic schedule and relatively low $\alpha_{\min}$ are optimal.

### 4.5.3 IS A SCHEDULE NECESSARY?

We investigate whether a schedule is necessary for good performance. This is equivalent to a "schedule" with the distribution alignment hyperparameter $k = 0$. We include a variety of settings for $\alpha_{\min}$

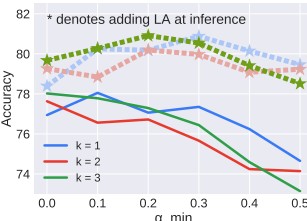 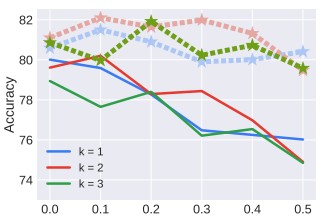 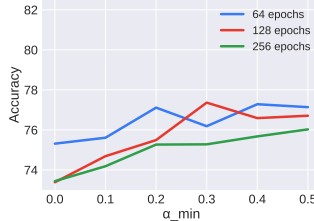

(a) **Evaluating a hybrid approach** which uses the same supervised loss changes as UDAL but with a form of progressive DA on the *pseudo-labels* rather than on the unsupervised loss.

(b) **Analysis of sensitivity to schedule hyperparameters**. We plot the classification performance of our method as a function of schedule parameters $k$ and $\alpha_{\min}$.

(c) **Removing the progressive nature of our alignment** shows a degradation in the performance of UDAL when $\alpha_{\min}$ is fixed for the entirety of training (i.e. $k = 0$).

Figure 3: A variety of ablations concerning the approach of our method.

and the duration of training in Figure 3c to ensure we adequately explore the trends in performance. We observe a significant loss in performance compared to the progressive version of UDAL presented in the paper. We attribute this to the fact that the *supervised branch* is immediately pushed to produce a marginal distribution of pseudo-labels that are more balanced. This, however, goes against empirical evidence (Kang et al., 2020) that representations are best learned from data sampled according to the data distribution, even if it is imbalanced.

## 5    RELATED WORK

Semi-supervised learning (Grandvalet et al., 2005; Lee, 2013; Laine & Aila, 2017; Berthelot et al., 2019; 2020; Sohn et al., 2020) has recently seen strong advances in performance. This can be attributed to the success of pseudo-labeling (Lee, 2013) combined with consistency of predictions (Berthelot et al., 2019; Sohn et al., 2020) among varying types of augmentations of unlabeled data.

While supervised learning has progressed significantly, a large amount of work has tackled the setting of class imbalanced learning (Kang et al., 2020; Jamal et al., 2020; Cui et al., 2019; Menon et al., 2021; Ren et al., 2020; Hong et al., 2021; Tang et al., 2020; Khan et al., 2017). These range from modifications to the loss formulation (Jamal et al., 2020; Menon et al., 2021; Ren et al., 2020; Hong et al., 2021; Khan et al., 2017) to decoupling representation from the classifier (Kang et al., 2020) and even modifying the optimization process itself (Tang et al., 2020).

By combining the two previous settings, we consider imbalanced, semi-supervised learning. Although still relatively unexplored, previous attempts (Hyun et al., 2020; Kim et al., 2020; Wei et al., 2021) to combat imbalance in the semi-supervised setting have been made. Both Hyun et al. (2020); Kim et al. (2020) modify changes to the loss formulation of a base semi-supervised learner to combat bias within majority classes, while Wei et al. (2021) involves a hybrid, generational approach that progressively aligns pseudo-labels predictions as well as augmenting the labeled set with rebalanced, confident pseudo-labels.

## 6    CONCLUSION

We presented Unifying Distribution Alignment as a Loss (UDAL) which addresses the issue of class-imbalance within semi-supervised learning. By connecting the ideas of progressive distribution alignment to logit adjustment, we provide a loss that can be applied to both the supervised and unsupervised branches rather than previous disjoint approaches. Furthermore, this approach incurs no additional training time on top of the underlying semi-supervised learner, achieves improved performance across multiple imbalanced settings and datasets, and scales to larger, more realistic datasets like ImageNet — all while requiring only a few lines of code. Future work includes applying UDAL in other settings, e.g. object detection where both imbalance and missing labels are pervasive.

## ETHICS STATEMENT

Our work is concerned with building more data-efficient classification models while reducing bias present in the distribution of data in the wild. While our work may be subject to similar claims of fairness as any supervised or partially supervised model, we do not believe it introduces harm and, if anything, attempts to mitigate effects that might be seen in biased datasets.

## REPRODUCIBLITY

Our work is based off of an open source release of CReST available here. The modifications we make are self-contained and Pythonic code is provided within the paper (e.g., Algorithm 1). All hyperparameters we introduce are documented within the paper.

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

# A   APPENDIX

## A.1   ALIGNMENT OF PSEUDO-LABELS OVER THE COURSE OF TRAINING

We examine the KL-divergence of the pseudo-labels (as a moving average) over the course of CIFAR10-LT training using UDAL.

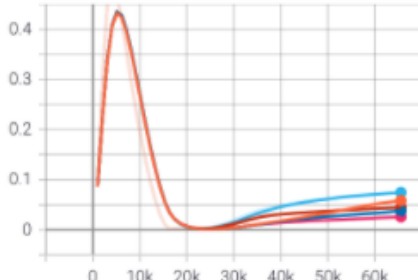

