# OpenReview forum: "Unifying Distribution Alignment as a Loss for Imbalanced Semi-supervised Learning"
_ICLR.cc/2022/Conference — ICLR 2022 Submitted_

### Official Review · Reviewer_inVm · 2021-10-25

**Correctness:** 3
**Technical Novelty And Significance:** 2
**Empirical Novelty And Significance:** 2
**Recommendation:** 3
**Confidence:** 4

**Main Review:**

## strengths
1. This paper studies an important problem which is underexplored in literature;
2. The proposed method is simple yet effective, which shows substantial performance gains on multiple datasets;
3. This paper is well written and easy to understand;
4. The proposed method is reasonable that aligns the distribution of both supervised and unsupervised losses.

## weaknesses
1. The novelty is limited. As mentioned, the proposed method is pretty much based on existing works, i.e., DA and LA, which are used and demonstrated to be helpful in class-imbalanced tasks. Importantly, the improvements of the methodologies are limited. For example, it is very natural to extend LA to unsupervised loss by using pseudo-labels. In this regard, this paper does not examine the quality of the generated pseudo-labels. Does the distribution of pseudo-label make sense? For the DA in Eq. (10), it immediately follows previous work, with a minor difference by introducing an additional parameter $k$;
2. More baselines should be compared. In the current version, this paper mainly compares its proposal with CReST+. I am aware that CReST+ is a strong baseline, but it would be more convincing if more baselines could be compared, such as [1,2,3];
3. Hyperparameters. From Figure 3, it is observed that the performance of the model is quite sensitive to the choice of $\alpha_{min}$;
4. In Table 4, the results for Supervised models are not informative. To better quantify the gap, LA or other techniques that handles class imbalance, should be also applied.


[1] DISTRIBUTION-AWARE SEMANTICS-ORIENTED PSEUDO-LABEL FOR IMBALANCED SEMI-SUPERVISED LEARNING.

[2] ABC: Auxiliary Balanced Classifier for Class-imbalanced Semi-supervised Learning. NeurIPS 2021.

[3] Unsupervised Semantic Aggregation and Deformable Template Matching for Semi-Supervised Learning. NeurIPS 2020.

**Summary Of The Paper:**

This paper studies class-imbalanced semi-supervised learning. To handle this problem, a unified approach is proposed by combing distribution alignment (DA) and logits adjustment (LA). In particular, this paper proposes to apply DA and LA to both supervised and unsupervised losses, which is new. This method shows significant improvement over baselines on three datasets.

**Summary Of The Review:**

My main concern is that the novelty is limited, which makes this paper like an incremental work. Therefore, I recommend rejection.

---

> ### Author Response · Authors · 2021-11-22
> **Response to Reviewer inVm**
>
> We appreciate the detail within your review.
>
> > **simple combination**
>
> We believe UDAL is not a simple combination (see "Novelty and contribution are limited" of out response to reviewer **qKfL**).
>
> It is unclear to us what "quality of the generated
> pseudo-labels" exactly means, however, we do keep track of the KL divergence between
> the pseudo-label distribution and labeled distribution (assumed to be
> equal) during training. We observe that over the course of training,
> these distributions to become quite close as UDAL attempts to align
> the pseudo-label distribution towards the labeled distribution. Finally, it
> starts to diverge from this near the end of training as UDAL attempts
> to align the pseudo-label distribution towards a more class-balanced one.
> We include this curve in the revised Appendix A1.
>
> > **More baselines**
>
> At the time of writing, we compare with all published work that we
> feel _confident_ is providing a fair comparison. Providing fair
> comparisons for semi-supervised learning is quite difficult unless all
> methods are faithfully implemented in the same codebase (see
> [Oliver et al](https://arxiv.org/abs/1804.09170)), comparisons are
> somewhat meaningless. Our framework implements both CReST+ (from their
> own release, which exactly replicates their paper numbers) and from DARP
> (using their dataset). We appreciate your recommendations of other
> works to compare with and we can attempt to compare with them. We note, however:
>
> **DASO**: This makes a strong assumption that USADTM is available. It is
> likely USADTM could be implemented as an additional component to
> improve any existing semi-supervised technique. Nonetheless*, in
> the main comparable setting within DASO (Table 1, CIFAR1-LT, $\gamma =
> 100$ aka our DARP setting), UDAL _significantly outperforms_ DASO 81.2
> (84.6 with LA) versus to 79.1 (82.1 with LA).
>
> **ABC**: While we think this is interesting work, we note that is is heavily
> concurrent work (having only been uploaded to arXiv after the ICLR
> deadline). Nonetheless*, in the main comparable setting where
> FixMatch is used as a base learner, UDAL significantly outpeforms it.
> For instance, on CIFAR10-LT:
>
> * $\gamma = 100, \beta = 10\%$. ABC: 77.2% verus UDAL: 80.2%
> * $\gamma = 100, \beta = 30\%$. ABC: 81.5% verus UDAL: 82.4%
>
> **USADTM**: While this is a significant work in semi-supervised learning (and is
> used within DASO), we believe this is targeted towards the case of
> balanced, semi-supervised learning and is not as relevant
> to our target problem.
>
> _Caveat_: Again, we emphasize this is assuming we can trust our
>   implementations of FixMatch are equitable to one another.
>
> > **hyperparameter $\alpha$**
>
> This is fair criticism. However, it is not
> uncommon to need to tune at least one hyperparameter in existing
> methods across disparate datasets (e.g. CReST on CIFAR versus
> ImageNet). We take some solace in the fact that UDAL has
> significantly faster convergence which might alleviate some overall
> tuning time requirements.
>
> > **Supervised models are not informative**
>
> Thank you, we will include updated accuracy numbers for the Supervised
> baseline. In particular:
>
>    * Supervised (10%) w/ LA: 54%
>    * Supervised (10%) w/ RandAug: 50%
>    * Supervised (10%) w/ RandAug + LA: 60%
>
> These still fall quite short of UDAL (75.3%) which almost approaches Supervised (100% performance).

---

### Official Review · Reviewer_qKfL · 2021-10-29

**Correctness:** 2
**Technical Novelty And Significance:** 2
**Empirical Novelty And Significance:** 2
**Recommendation:** 3
**Confidence:** 2

**Main Review:**

This paper focuses on class-imbalanced semi-supervised learning and proposes a new method that combines distribution alignment and logit adjustment techniques. The proposal is simple. However, the proposed method relies on the assumption that the imbalance distribution between labeled and unlabeled datasets is the same. This is a very strong prior knowledge and in reality, we can not obtain this knowledge.  Thus, the application of the proposal is limited. Moreover, the new proposal is mainly based on existing techniques, although combining these techniques can achieve a performance improvement, the novelty and contribution are limited.

**Summary Of The Paper:**

This paper focuses on class-imbalanced semi-supervised learning and proposes a new method that combines distribution alignment and logit adjustment techniques. Experiments show that the proposal can achieve performance improvement.

**Summary Of The Review:**

This paper proposes a new method for class-imbalanced semi-supervised learning. However, in my view, the assumption of the proposal is too strong to satisfy and the novelty of the proposal is limited.

---

> ### Author Response · Authors · 2021-11-22
> **Response to Reviewer qKfL**
>
> > **Proposal is too strong**
>
> We note that this is the _de facto_ assumption so far in our field with only a small amount of
> attention to the case where they might be unequal. We believe that our
> assumptions are in line with significant works in the field:
>
> 1. [DARP](https://arxiv.org/abs/2007.08844) (NeurIPS 2020) which makes
> this same assumption for their primary results (see Table 1)
> 2. [CReST](https://openaccess.thecvf.com/content/CVPR2021/papers/Wei_CReST_A_Class-Rebalancing_Self-Training_Framework_for_Imbalanced_Semi-Supervised_Learning_CVPR_2021_paper.pdf)
> (CVPR 2021) which makes this same assumption for all results.
> 3. [ABC](https://arxiv.org/pdf/2110.10368.pdf) (concurrent work,
>    uploaded to arXiv post-ICLR deadline, NeurIPS 2021) which makes
>    this same assumption for all results.
>
> Therefore, most recently published work is based off of the exact same
> assumptions we make in our work. Furthermore, we believe that there is
> a reasonable statistical argument to be made supporting this
> assumption. Suppose one gathers a large amount of unlabeled data which
> they know comes from some set of $C$ classes. Then, annotating a small
> subset and treating this as a labeled set for semi-supervised learning
> would treat this as a sample in a larger estimation problem of the
> marginal class distribution of the entire set. Thus, statistically,
> the marginal distribution of our labeled set will approximate the
> marginal distribution of the unlabeled set with some error decreasing
> as the size of the labeled set increases.
>
> **We examine estimating the unlabeled distribution parameters in the response to Reviewer aN2N**.
>
> > **Novelty and contribution are limited**
>
> While we appreciate that UDAL can be viewed as a simple method (and
> despite this, outperform other more complicated methods), we don't
> believe it is a ``simple combination''. We outline what we would
> consider naive baselines that more readily qualify as simple
> combinations and show that these simply do not work:
>
> 1. Apply LA to supervised branch (using the fixed labeled
>    distribution) and apply progressive DA to the unsupervised branch:
>
> This does not work well because applying LA to the supervised branch
> without consideration of the impact this might have on the
> _unsupervised_ branch will lead to a strong class re-balancing effect
> on pseudo-labels from the beginning of training and will yield poor
> results. We gave an example of this suboptimality in the 3rd row of
> the FixMatch section in Table 1 (w DA ($\alpha_{min} = 0.5)$) to show
> a drop from 80.2 (UDAL) to 73.6 (this baseline) under the $gamma =
> 100$ setting.
>
> Therefore, in the paper, we must use a progressive form of LA (which
> has not been considered before). This adaptively applies LA towards a
> target distribution ($\tilde{p}_{\alpha_t}$) that matches the
> labeled distribution at the beginning of training and towards a more
> balanced distribution at the end. Therefore, we can ask what might
> occur when we apply this progressive form of LA and the (existing)
> progressive form of DA.
>
> 2. Applying progressive LA to the supervised branch and progressive DA
>    to the unsupervised branch.
>
> This is a great baseline which _actually performs well_. We provide a
> sweep of results across a variety of hyperparameter settings to
> understand this:
>
> ### Accuracy under CIFAR 10-LT ($\gamma = 100$)
>
> * UDAL: 80.2%
>
> * Best 5 settings for Progressive LA (Supervised) + Progressive DA
> (Unsupervised) across sweep of $k$ and $\alpha_{min}$
>
> | $\alpha_{min}$ | $k$ | Accuracy |
> | -------------- | --- | -------- |
> | 0.0            | 3   | 77.9     |
> | 0.1            | 1   | 77.9     |
> | 0.2            | 2   | 77.6     |
> | 0.1            | 1   | 77.5     |
> | 0.1            | 2   | 77.5     |
>
> We will make sure to include this in the final paper.
> However, even in the best settings, we still observe a sizable gap in
> performance between UDAL and ours. Our hypothesis is that while these
> two mechanisms: LA and DA are similar in spirit, they are very
> different in mechanism:
>
> 1. LA adjusts the _loss_ computation which indirectly affects the bias of the classifier
> 2. DA adjusts the _pseudo-label distribution_ computation which must then go through some thresholding
> to produce PLs (and thus affect the classifier).
>
> Therefore, using these two methods provides a disparate manner in which imbalance is approached. Empirically,
> it seems to affect the overall upper bound in improvement in performance.
>
> _Hence_, the idea of applying LA for _both_ branches (rather than LA for one and DA for the other)
> is the logical premise of our paper and we believe it has sufficient novelty,
> especially when coupled with good performance. In order to do this, we needed to connect DA to LA which
> has so far been overlooked. Altogether, by applying LA to both branches against the same target distribution
> is what we believe allows UDAL to perform as well as it does.

---

### Official Review · Reviewer_aN2N · 2021-11-02

**Correctness:** 4
**Technical Novelty And Significance:** 3
**Empirical Novelty And Significance:** 3
**Recommendation:** 6
**Confidence:** 4

**Main Review:**

Strengths:
+ This paper addresses the topic of semi-supervised learning in cases where the underlying data distribution is severely imbalanced. This is an underexplored problem that is relevant to the AI/ML community and has practical real-world impact.
+ In contrast to other papers addressing this problem, the proposed approach relies only on diistribution alignment without sampling-based strategies. This is done by moving the distribution into the cross-entropy loss computations of recent semi-supervised learning algorithms: FixMatch and MixMatch.
+ This work connects ideas of progressive distribution alignment with logit adjustment from the fully supervised, imbalanced learning setting.
+ By considering distribution alignment alone, training time is reduced (5x reduction from CReST).
+ The method achieves significantly better performance characteristics as more labeled data becomes available, and the approach scales to larger datasets, outperforming the best existing method on ImageNet-127 (in terms of accuracy).
+ The proposed method is well-framed in the existing literature.
+ Section 2 provides a nice overview of the prerequisities needed to understand the method.
+ Eqs. 8 and 9 are interesting insights.
+ The approach is simple, requiring only a few additional lines of code to implement with no alteration to the training scheme and no increased training time.
+ The approach only has two hyperparameters which need to be tuned.
+ Experiments are conducted over three benchmark vision datasets of varying complexity: long-tailed versions of CIFAR10, CIFAR100, and ImageNet-127.
+ Experimental setup seems reasonable, and follows a similar procedure to CReST.
+ Experimental results are competitive or exceeds other methods on the tested datasets.
+ Ablations are run to validate each component of the proposed approach.

Weaknesses:
- Imbalanced semi-supervised learning, while an important problem, is not a new problem, somewhat limiting novelty.
- The proposed approach is a combination of two existing approaches, somewhat limiting novelty.
- Experiments are conducted on only a single network architecture. It would be useful to show it works on a wide range of neural network architectures.
- It would be useful to see if the results are statistically significant.

Questions:
- What happens if the assumption that the unlabeled set of examples is imbalanced in a different way than the training examples?
- Please define "strong" and "weakly" augmented versions of the unlabeled example.
- Are there any significant drawbacks to this approach compared to others?
- What happens in the balanced setting; does it fall back to a standard semi-supervised learning approach?

**Summary Of The Paper:**

This paper addresses the topic of semi-supervised learning in cases where the underlying data distribution is severely imbalanced. This approach combines distribution alignment with logit adjustment, resulting in an efficient method for solving the aforementioned problem while improving performance in the test setting. Unlike existing state-of-the-art approaches such as CReST, the approach involves no sampling state, and imbalance mitigation is achieved just by modifying the loss functions of the model. Experiments are conducted over three benchmark vision datasets of varying complexity: long-tailed versions of CIFAR10, CIFAR100, and ImageNet-127. Experimental results are competitive with or exceed other methods on the tested datasets with 5x training speedup compared to CReST.

**Summary Of The Review:**

This paper addresses the topic of semi-supervised learning in cases where the underlying data distribution is severely imbalanced. This is an underexplored problem that is relevant to the AI/ML community and has practical real-world impact. In contrast to other papers addressing this problem, the proposed approach relies only on diistribution alignment without sampling-based strategies. This is done by moving the distribution into the cross-entropy loss computations of recent semi-supervised learning algorithms: FixMatch and MixMatch. The paper is generally well-structured and clearly written. The experimental set up is sound, and experimental results show the promise of the model. The approach is well-grounded in the existing literature, and its motivation is clear. As it stands, I do not see any major flaws with the approach.

---

> ### Author Response · Authors · 2021-11-22
> **Response to Reviewer aN2N**
>
> Thank you for your generally positive outlook
>
> _Per weaknesses:_
>
> > **while an important problem, is not a new problem**
>
> We believe imbalanced, semi-supervised learning (as applied to
> image classification) is still somewhat new. However, we claim no
> novelty in proposing the setting, but rather a novel, heavily
> simplified approach that outperforms previous approaches, while
> providing some connections between techniques in the mitigations for
> imbalance between the supervised and unsupervised literature.
>
> > **is a combination of two existing approaches**
>
> Please see _Novelty and contribution are limited_ in the response to **Reviewer qKfL**.
>
> > **single network architecture**
>
> This is true, however, this is commonplace in ensuring standard
> architectures are used across papers. We do note, however, that
> ImageNet results are computed on a standard ResNet architecture
> (versus the WRN architecture in earlier experiments) which does show
> some range in applicability.
>
> _Per questions:_
>
> > **What happens if the assumption**
>
> Please see the response to **Reviewer WjMc**: _in the case of an unknown_
>
> > **Please define**
>
> These are covered in the
> [FixMatch](https://arxiv.org/abs/2001.07685) paper. As a summary,
> strong augmentations provide a heavy amount of distortion
> (e.g. AutoAugment, RandAugment, Cutout) while weak augmentations are
> generally benign e.g. flip or no augmentation at all. We will elaborate on these in an updated version.
>
> > **significant drawbacks**
>
> We believe the only real drawback is the need to possibly tune
> $\alpha_{min}$ across disparate datasets (e.g. CIFAR to
> ImageNet). However, this is often the case for existing methods which
> require some hyperparameter tuning (we only require one). Nonetheless,
> UDAL converges faster than most other methods and so this tuning might
> not be that significant compared to other methods.
>
> > **balanced setting**
>
> It does fallback to exactly FixMatch in the balanced case. In this
> case, $p_{data}$ will be uniform and thus dividing by it will not
> affect the computed losses throughout training, causing them to
> fallback to standard FixMatch training.

---

> > ### Comment · Reviewer_aN2N · 2021-11-28
> > **Thank you for your responses**
> >
> > Thank you for your responses; I think they adequately address my concerns. Based on the other reviewers' concerns w.r.t. novelty, I have decided to slightly reduce my score, but I still think the paper is above the acceptance threshold.

---

### Official Review · Reviewer_WjMc · 2021-11-02

**Correctness:** 4
**Technical Novelty And Significance:** 3
**Empirical Novelty And Significance:** 3
**Recommendation:** 6
**Confidence:** 4

**Details Of Ethics Concerns:**

None.

**Main Review:**

**Pros.**

- **Clarity**. Overall, the writing is clear and easy to follow.
- **Important problem with a simple and effective solution**. Considering the class imbalance scenario is an essential step for applying SSL in a more realistic scenario but has been less explored. The proposed method can be used with a simple modification of the existing baseline and it shows significant empirical gain; hence, it has the potential to be widely used to mitigate this problem without additional burdens. Although the proposed method can be viewed as a simple combination of the existing methods, I believe that the simplicity and effectiveness of the proposed method will be interesting to the reader in ICLR.

**Cons.**

- **More general imbalanced semi-supervised learning scenarios.** Although I agree that a considered scenario (i.e., distributions of both labeled and unlabeled are same) is most natural, I wonder that the proposed method can be applicable to more generic scenarios (i.e.,  distributions of both labeled and unlabeled are different). Is there any way to apply the proposed UDAL to such a challenging scenario?
- **Effect of the strategy in (Kang et al., 2020).** In Section 4.3., the authors present that they use a deferred re-sampling strategy which is introduced by Kang et al., 2020 ("... spend the very last stages of training aligning to a relatively balanced class distribution..") Can the authors give the details about this? Also, is this necessary for empirical improvement?

Kang et al., Decoupling representation and classifier for long-tailed recognition., In *ICLR*, 2020

**Summary Of The Paper:**

- This paper tackles the class-imbalanced problem with a semi-supervised learning scenario. Unlike the previous approaches, which require a complicated sampling strategy and multiple training pipelines, the authors provide a simple and unified framework, UDAL, by connecting the ideas of progressive distribution alignment (proposed for imbalanced semi-sup) to logits adjustment (proposed for imbalanced sup). This approach incurs no additional training time on top of the underlying semi-supervised learner. Significant empirical improvement on widely used benchmarks (CIFAR-10-LT, CIFAR-100-LT, and ImageNet-127) demonstrates the effectiveness of the proposed method.

**Summary Of The Review:**

Although the proposed UDAL can be viewed as a simple combination of the existing methods (progressive distribution alignment and logits adjustment), I believe that the simplicity and empirical effectiveness of the proposed method will be interesting to the reader in ICLR.

---

> ### Author Response · Authors · 2021-11-22
> **Response to Reviewer WjMc**
>
> Thank you for your generally positive outlook. Per your questions:
>
> > **More general imbalanced semi-supervised learning scenarios**
>
> ## In the case of an unknown (and unequal) unlabeled class distribution:
>
> As noted in the DARP paper, even in this case, we can still estimate
> the unlabeled marginal distribution using confusion matrices over a
> small hold-out set trained on the labeled data. We use this technique
> to estimate when the imbalance ratios might not be the same. We assume
> the relative ordering of classes is the same (the most frequent and
> least frequent classes are the same across each set).
>
> * $\lambda_l$ denotes the imbalance ratio of the labeled set
> * $\lambda_u$ denotes the imbalance ratio of the unlabeled set
>
> We compare the estimated scenario to the upper bound when these
> factors are known. In either case, we use UDAL to align the supervised
> branch to the distribution given by $\lambda_l$ and use $\lambda_u$ or
> the estimated distribution using the DARP technique in order to align
> the unsupervised branch. Results are on CIFAR10-LT (DARP).
>
> | $\lambda_l$ | $\lambda_u$ | Accuracy (Known) | Accuracy (Estimated)
> | -------------- | --- | -------- | ------ |
> | 50             | 100 | 80.1     | 79.7  |
> | 100            | 150   | 82.9     | 82.4  |
>
> While _knowing_ each distribution performs best, we find accuracy only
> suffers in a minor way when UDAL is used with the estimated unlabeled
> distribution. Furthermore, UDAL still works when these distributions
> are known but perhaps unequal. From the paper, we only see a drop from
> 86.5 ($\lambda_l = \lambda_u = 50$) to 82.9 ($\lambda_l = 50$,
> $\lambda_u = 150$) and from 81.4 ($\lambda_l = \lambda_u = 100$) to
> 80.1 ($\lambda_l = 100$, $\lambda_u = 150$). Some drop is to expected
> since tail classes are less frequent within the unlabeled data at
> higher imbalance ratios.
>
> > **Effect of the strategy in (Kang et al., 2020)**
>
> Kang et al., 2020 consider imbalanced,
>    supervised learning. To mitigate issues related to imbalance, they
>    decide upon a fixed strategy of training against the data
>    distribution for the bulk of training. After this, they
>    reinitialize the classifier (last linear layer) and re-train this
>    using a class-balanced sampler. This is important because sampling
>    against the data distribution learns a good representation, but
>    class-balanced sampling learns a balanced classifier. On the other
>    hand, we clarify that our approach uses no sampling, however, our
>    alignment strategy also considers a “target” distribution ($\tilde{p}_{\alpha^t}$) which
>    shifts over the course of training. Like Kang et al., 2020, we want
>    this target to be close to the data distribution at the beginning
>    of training and close to the class-balanced distribution at the
>    end. However, unlike Kang et al., 2020, we interpolate between data
>    distribution and class-balanced distribution over the course of
>    training rather than a discontinuous jump between these modes. This
>    progressive interpolation is entirely necesssary for our results
>    based off of our experience.
>
> > **UDAL is a simple combination**
>
> Please see the response to **Reviewer qKfL**: _Novelty and contribution are limited_ for why we feel that UDAL is not only a simple combination.

---

> > ### Comment · Reviewer_WjMc · 2021-11-24
> > **After Rebuttal**
> >
> > Thank you very much for the response. I appreciate the effort that the authors put into addressing my questions. I believe that the above results and discussion can significantly improve the quality of the manuscript. My major concerns are mostly addressed; however, I also agree with the other reviewers' concerns. Hence, I'll keep my score (**6**).

---

> > > ### Author Response · Authors · 2021-11-27
> > > **Thanks for your response**
> > >
> > > Dear Reviewer WjMc,
> > >
> > > We thank you for reading our rebuttal and finding our response addressing most of your concerns. We also addressed other reviewers’ concerns point-by-point in the rebuttal. We would be happy to answer any partially addressed issues from other reviewers and welcome the guidance on further clarification.
> > >
> > > Thank you \
> > > Authors

---

### Decision · Program_Chairs · 2022-01-20

**Decision:**

Reject

**Comment:**

Thanks for your submission to ICLR.

Reviews were fairly mixed on this paper, with two reviewers advocating for accepting the paper and two advocating for rejecting the paper.  There were some concerns raised by the reviewers, most notably novelty and some issues with the experiments.  After rebuttal, the negative reviewers maintained their scores and the positive reviewers were somewhat less enthusiastic.  In the end, the paper is quite borderline and could really go either way, but it seems that the paper could use another round of reviewing, particularly to make sure the issues raised by the reviewers are adequately addressed.

Please do keep in mind their comments when preparing a future version of the manuscript.